# Towards the Implementation of a Conceptual Framework of Food and Nutrition Literacy: Providing Healthy Eating for the Population

**DOI:** 10.3390/ijerph16245041

**Published:** 2019-12-11

**Authors:** Virginia Vettori, Chiara Lorini, Chiara Milani, Guglielmo Bonaccorsi

**Affiliations:** 1Department of Health Sciences, University of Florence, 50134 Florence, Italy; chiara.lorini@unifi.it (C.L.); guglielmo.bonaccorsi@unifi.it (G.B.); 2School of Specialization in Hygiene and Preventive Medicine, University of Florence, 50134 Florence, Italy; chiara.milani@unifi.it

**Keywords:** food literacy, nutrition (or nutritional) literacy, scoping review, executive framework

## Abstract

Existing definitions of food literacy (FL) and nutrition literacy (NL) in particular refer to individual knowledge, motivation, competences, and awareness, which determine the relationship between individuals and food, the food system, and nutrition information. Several authors proposed specific conceptualization of the terms. Nevertheless, the description of analogies and differences between FL and NL is still lacking, as is an integrated framework which highlights the meaning of the concepts. This work aims to describe and discuss evidence provided by the literature in order to develop and propose a comprehensive conceptualization of FL and NL to the scientific community. We systematically reviewed six databases, considering the search terms of FL and NL. We collected the antecedents, components, and consequences of both FL and NL. We underlined and traced similarities of the concepts as well as prerogative features through the content analysis of definitions. We obtained 14 definitions of NL and 12 definitions of FL; 42 papers presented antecedents and 53 papers contained consequences. We observed that NL could be considered a specific form of FL. In addition, we noted that the consequences of NL are included in the subset of the consequences of FL and the conceptual limits of FL correspond to the outcome of healthful diet. We conclude that FL and NL build a multifaceted concept which implies both individual and public perspectives. We propose a conceptualization which could be useful to develop an executive framework aimed at providing healthy eating for the population.

## 1. Introduction

Food and nutrition literacy are relevant issues in achieving the sustainability of the food system, which has an important impact on public health and environmental health [1]. More specifically, they may play a vital role in improving individuals’ eating quality.

The concepts of food literacy (FL) and nutrition literacy (NL) have been defined only recently because of their role in human health. For a long time, our diet simply followed the human ability to hunt, grow, and breed. In the current society, consumers are becoming increasingly disconnected from nature and the foods provided despite the big interest in foodstuffs and diets [2]. The concepts of FL and NL probably originate from that of health literacy (HL) [3], as a result of individuals’ need to orient themselves in a complex food environment through specific knowledge and competences.

The term FL appeared for the first time in a book published in 1992, providing nutritional advice and recipes from a cooking professional [4]. Shortly afterwards, the first definitions of FL and NL appeared in the literature in 1995 and in 2001, respectively [5,6]. To date, there are several publications that concern FL and NL, with the majority being published in Canada, Australia, and the United States [7]. The oldest definitions of the terms mostly adopt an individual perspective involving a limited conceptualization of the topic. Subsequently, some definitions defined FL and NL referring to Nutbeam’s tripartite model [8,9], which considers three literacy levels: functional, interactive, and critical. Functional literacy includes declarative and procedural knowledge to obtain information and awareness regarding facts and processes, as well as practical skills and strategies. In interactive literacy, the interactions between individuals and opportunities to exchange, share, and discuss information and participate in shared actions are considered. The critical level focuses on a critical analysis of information and understanding the food impact on the environmental and socio-economic fields. Additionally, it includes actions addressing barriers to human nutritional health and the sustainability of the food production system.

In 2014, based on the perspectives of both experts and young people, Vidgen and Gallegos [10] developed their idea of FL, proposing that it involves a wide range of knowledge, skills, and behaviors that provide individuals with the capacity to preserve diet quality. According to Vidgen and Gallegos [10], the authors of six papers expanded the perspective towards the health dimension, including the promotion of nutritional health and a sustainable food system through individuals’ food choices [11,12,13,14,15,16]. The authors of seven previous papers emphasized the potential role of FL and NL as tools to achieve both population health and environmental outcomes.

Through this literature review, we attempted to grasp and synthesize the comprehensive meaning of FL and NL and their relationship by systematically identifying and collecting their definitions as well as their antecedents and consequences. We also conducted a content analysis of the definitions collected. 

This study aimed to provide a solid basis for the definition of FL and NL from the perspective of public health, considering individual, collective, and contextual issues. 

## 2. Materials and Methods

We conducted a literature review focusing on the meaning of the terms ‘food literacy’ and ‘nutrition (or nutritional) literacy’. We used the methodological approach of Sørensen [17], in which HL was conceptualized in a comprehensive framework including its antecedents and consequences. Regarding the antecedents, we based this on the classical model of social determinants of health [18]. We considered distal (socio-economic conditions, cultural, and environmental conditions, education, living and working conditions, housing) as well as more proximal (age, sex, and general literacy) factors which could potentially influence individuals’ FL and NL. Regarding the consequences, we reviewed papers considering health-related outcomes (i.e., body mass index and lipid blood concentration), and other factors that may impact individuals’ health (i.e., food safety and food security).

### 2.1. Search Strategy

We conducted a scoping review in line with the methodology and guidance for the development of systematic scoping reviews outlined and developed by members of the Joanna Briggs Institute and members of five Joanna Briggs Collaborating Centres [19]. We explored six databases: Medline (PubMed), Embase, Web of Science, Cochrane library, Health Evidence, and Trip database. We used the following search string: ‘food literacy’ OR (‘nutrition literacy’ OR ‘nutritional literacy’) (See Figure 1).

In accordance with Krause et al. [3], Truman et al. [7], Palumbo [16], and Azevedo Perry [20], we decided to apply the simple combination of keywords with the purpose of identifying all definitions and conceptualizations of the topics. 

### 2.2. Inclusion Criteria

Considering the purpose of the review, no temporal limit was applied, and only articles published in the English language were considered. Databases were searched until 6 April 2018. Abstracts, theses, reports, conferences, books, and web sites were excluded; however, we included scoping reviews, narrative reviews, conceptual papers, and editorials, as they could be useful to identify definitions of the terms or explore the meaning of the concepts. After reviewing titles and abstracts, the articles that did not cover the topic of interest were removed. We also excluded studies for which we did not find either full text or abstract, ongoing studies, and duplicates derived from different databases. Of the 349 articles identified through the database search, 125 met the inclusion criteria. 

### 2.3. Study Selection

We selected the articles that focused on the topic of FL or NL by the screening of the objectives of each study. Studies were selected when they explored the meaning of FL or NL, or defined them, identified health-related outcomes, described programs or projects related to FL or NL, or developed or evaluated specific measurement tools. 85 articles were eligible to review. The systematic search of the literature using the search string was independently performed by two authors (V.V. and C.M.). The same authors (V.V. and C.M.) screened the full text of the 85 papers and collected definitions, antecedents, and consequences.

### 2.4. Data Extraction

We reviewed the full text of the articles. We developed a summary table to record the characteristics of the included studies and the key information relevant to the research question in line with the guidance regarding how to conduct a scoping review [19]. We extracted, summarized, and tabulated the following key information from each publication: title of the publication, author(s), year of publication, country of origin, abstract of paper, definition of FL or NL (if available), antecedents and consequences (if available). 

First, we examined all the articles for definitions (for a detailed overview of the data extraction of definitions See Appendix A). Second, we reviewed all the articles to identify antecedents and consequences of FL and NL. Regarding the definitions, we considered only statements that explained the meaning of the terms proposed by the authors, introduced by words such as ‘is’, ‘may/can be defined’, or ‘defined/described as’ (i.e., ‘Food literacy is about acquiring and developing the food-related skills necessary to help create behavior change [21] (p. 342)’ or ‘Nutrition literacy may be defined as the degree to which people have the ability to obtain, process, and understand basic diet information and the tools needed to make appropriate nutrition decisions [22] (p. 422)’.

As for antecedents and consequences, we checked the sentences that described predictors of FL and NL (i.e., ‘a number of factors are suggested to be associated with the decline and devaluing of food literacy components such as cooking within the population [23] (p. 158)’; ‘cognitive difficulty was inversely related to nutrition literacy [22] (pp. 427–428)’) and the outcomes of FL and NL (i.e., ‘food literacy may influence adolescents’ dietary intake [24] (p. 824)’; ‘a significant relationship were found between five of the six domains of nutrition literacy and diet quality [25] (p. 493)’). We did not include any additional records identified through other sources (e.g., reference lists) because we considered the list of the studies collected sufficiently wide for the purpose of this research.

### 2.5. Data Analysis and Synthesis

Even though Krause et al. [3] studied FL and NL definitions applying Nutbeam’s model, we preferred to consider seven specific clusters to analyze FL and NL definitions for the purpose of identifying analogies and differences between the concepts.

To conduct content analysis of definitions, we developed an analytic grid considering seven clusters of features: knowledge, competence, skills, and awareness: (K/C/S/A), actions (A), information and resources (I/R), subject (S), objective (O), context (C), and time (T) (See Table 1). We considered the same clusters identified by Sørensen et al. [17] in their analysis of HL definitions except one (subject (S)) that we added with the aim to distinguish whether the definition was referred to an individual or a wider perspective.

One researcher (V.V.) conducted the content analysis and examined FL and NL definitions according to each cluster. Sentences, part of these, or a single word of the definitions related to the clusters were underlined and tabulated. Based on the summary table, two reviewers (G.B. and C.L.) independently reviewed the analysis of the definitions. The specific characteristics of the constructs were condensed for each cluster. The content analysis of definitions allowed us to comprehensively understand the constructs of FL and NL. 

Regarding antecedents and consequences, key sentences were collected and discussed by the research team, resulting in a list of topics that traced the boundaries of FL and NL. Specifically, based on the meaning of the antecedents and consequences of FL and NL, two authors (V.V. and C.M.) reviewed the papers and underlined key sentences reporting the antecedents or consequences. Secondly, all the sentences were tabulated in an electronic spreadsheet, which produced a summary table recording the characteristics and key information of the included studies.

## 3. Results

### 3.1. Definitions

#### 3.1.1. Definitions of NL

We found 14 definitions of NL, which generally described knowledge, skills, and competence necessary for nutritional health (for a detailed overview of the definitions, see Appendix A).

Five papers directly referred to the classical definition of HL [26] and defined NL as the degree to which individuals are able to obtain, process, and understand nutrition and diet information, as well as access services needed to make adequate nutrition decisions [22,27,28,29,30]. In addition to basic literacy skills (reading and writing), three papers [5,14,31] emphasized basic quantitative skills that are necessary to understand concepts of healthful diets and information regarding nutrition. The ability to read and comprehend food labelling requires both literacy and numeracy. Guttersrud et al. [14] and Escott-Stump [31] included this theme in their NL definitions, and Sullivan and Gottschall-Pass [5] illustrated the characteristics of ‘label nutrition literacy’, the ability to use food labels. 

Two definitions [13,14] expanded the concept of NL directly referring to Nutbeam’s tripartite model [8,9]. On the one hand, Doustmohammadian et al. [13] and Guttersrud et al. [14] included typical functional characteristic elements, such as reading and writing, necessary to understand and follow nutrition messages and grasp the essence of nutrition indications. On the other hand, in their definitions, they considered an advanced level of literacy (interactive literacy), which includes cognitive and interpersonal skills needed to interact adequately with nutrition counsellors, experts, or others (e.g., relatives). Furthermore, they focused on the ability to critically analyze nutrition information and advice, and to engage in actions to address barriers to individual and global nutritional health (critical literacy). Gibbs et al. [32] also referred to the critical level of NL, defining the ability to navigate nutrition-related information. Palumbo [16], Liao and Lai [33], Lee et al. [34], and Cassar et al. [35] summarized the concept of NL, defining it as the capacity or knowledge and skills required for the selection of a healthy diet in everyday life.

#### 3.1.2. Definitions of FL

We retrieved 12 definitions of FL (for a detailed overview of the definitions See Appendix A). Generally, FL definitions included both knowledge and skills related to food and individual’s awareness, behaviors, and actions. Kolasa et al. [6] developed the FL construct as three actions, namely obtain, interpret/understand, and use, which determine the access to and use of food information and services. This conceptualization was based on the construct of HL developed by Nutbeam [8]. Block et al. [11] expanded this point of view by suggesting that FL is more than knowledge and involves the empowerment of individuals, and they proposed the idea of ‘food well-being’. The achievement of global nutritional health is gained through the understanding of food and nutrition and acting on that knowledge [11]. Page-Revees et al. [36] focused on the critical dimension of Nutbeam’s model [8,9] and emphasized the concept of empowerment, and they described Hispanic women’s everyday experiences about food insecurity. The first step to reach food security is to be aware of the impact that the food system plays on households [36]. Vidgen and Gallegos [10] symbolized FL as a ‘scaffolding’ that protects diet quality through adequate individuals’ food choices. This definition strongly involves the concept of empowerment as well. According to their point of view, FL includes the knowledge and awareness necessary to use food in order to ensure nutritional health and a healthy diet. Thus, FL includes diverse knowledge, skills, practices, and behaviors [10]. 

The FL definition of Cullen et al. [12] involves a multifaceted dimension. On the one hand, the authors of this paper emphasized the importance of developing a positive relationship with food and being able to navigate the complex food system, which involves food skills and practices developed across the lifespan. On the other hand, they focused on individuals’ capacity to take health enhancing actions aimed at improving nutritional health and achieving a sustainable food system. These characteristics of the FL construct recurred in other definitions proposed in the reviewed literature [15,16].

Some other definitions of FL focused on individuals’ relationship with food [21,37,38,39,40]. Five previous papers agreed that FL represents a set of skills, knowledge, awareness, and behaviors that allows individuals to adequately interact with food, from being able to prepare it in order to meet nutrition guidelines, up to navigating the complex food system. Finally, Slater [41] referred to Nutbeam’s model [8,9] considering three different levels to deal with food and nutrition issues. On the first level, the author included communication, understanding, and use of food and nutrition information. On the second level, the author conceived skills including decision making, goal setting, and practices to achieve nutritional health and a state of well-being. Finally, considering different cultural, family, and religious beliefs related to food and nutritional issues is characteristic of this conceptualization.

### 3.2. Comparison between NL and FL

We detected that the core elements of NL and FL substantially differed. NL definitions mostly involved nutritional information and individuals’ capacity and interest in relation to accessing and using such information [5,27,28,29,30,31,32] in order to maintain nutritional health [13,14,22]. Few definitions directly referred to the ability to select a healthy diet [16,33,34,35], offering a more synthesized conceptualization of the dimension.

On the other hand, most FL definitions considered food and the capacity to use it and interact with it as the core element of the construct [10,12,16,21,36,37,38,39].

The content analysis of definitions is outlined in Table 2. 

#### 3.2.1. Knowledge, competence, skills, and awareness (K/C/S/A)

NL definitions emphasized not only individuals’ ability to access and use information but also the degree to which they are able to do this [14,22,28,30]. In addition, specific types of skills characterized NL construct. Quantitative skills are useful to access and understand food labels and menu labelling information [5,14,31]. Cognitive and interpersonal communication skills enable adequately interacting with experts [13,14,31]. Palumbo [16], Liao and Lai [33], Lee et al. [34], and Cassar et al. [35] synthesized the NL concept to a greater degree and explained it as individuals’ ability or knowledge aimed at choosing a healthful diet.

Regarding FL definitions, six papers conceived the concept as individuals’ immaterial scaffolding, ability, or awareness aimed at accessing and correctly using food [10,12,16,21,38,39].

#### 3.2.2. Actions (A)

Some verbs (i.e., to grasp the essence, to follow, to problem solve, to make right food choices, to interact with) could be found exclusively when reviewing NL definitions [5,13,14,31,35]. We also noted that the actions that composed both NL and FL constructs generally referred to the three verbs (access, understand, use) of the HL definition by Nutbeam [8].

Some other verbs were exclusive to the FL construct and referred to individuals’ relationship with food (i.e., to select/purchase/prepare/preserve, to navigate/engage/participate, to support, to respect, to protect, to advocate) [10,12,15,37,39,41]. Finally, the concept of empowerment as the actions following from individuals’ food knowledge and awareness often recurred when reviewing FL conceptualizations [10,11,36].

#### 3.2.3. Information and Resources (I/R)

Both NL and FL constructs included nutritional information, but the theme of individuals’ connection with food and the food system was discussed only in FL definitions. On the other hand, NL definitions explained more in depth the kind of information that literate individuals need to deal with. Actually, Guttersrud et al. [14] and Palumbo [16] talked about individuals’ manner of dealing with nutrition information guidelines and advice, and the concept of healthful diets; Sullivan and Gottschall-Pass [5], Guttersrud et al. [14], and Escott-Stump [31] focused on information on front-of-package, food labels, and menu labelling; and the authors of eight papers discussed nutrition information or issues, and basic nutrition information [13,14,22,27,28,29,30,32].

#### 3.2.4. Subject, Context, and Time (S, C, T)

In terms of the subject, Guttersrud et al. [14] described the NL construct moving from a personal to a population perspective. Regarding the context, Doustmohammadian et al. [13], Guttersrud et al. [14], and Escott-Stump [31] described the nutrition environment as a situation in which a person interacts and spends time with a nutritionist or expert. The authors of seven papers discussing FL adopted both individual [6,12,16] and wider perspectives [10,36,38,41]. Additionally, Vidgen and Gallegos [10] and Cullen et al. [12] described individuals’ FL as developing across the lifespan.

#### 3.2.5. Objective (O)

Further, in this case, analogies and differences arose when comparing the two constructs. When either FL or NL are achieved, individuals’ nutritional health can be enhanced. However, the accomplishment of a sustainable food system represents a prerogative of the FL construct [12,15,16,38].

### 3.3. Antecedents of NL and FL

We identified 42 papers dealing with the antecedents of NL and FL (See Figure 2). Figure 2 combines the antecedents and consequences of NL and FL.

The role of some antecedents and consequences discussed in the text was demonstrated by primary studies [25,42,43,44,45].

#### 3.3.1. Antecedents of NL

As for NL antecedents, the authors of seven papers [22,25,31,35,45,46,47] observed that professionals in dietetic and nutritional fields have a key role in positively influencing NL, since they disseminate evidence-based information to patients during doctor appointments [46]. Considering the youngest subjects, the role of the expert in imparting adequate knowledge and practices is taken over by parents. The role of parents can be considered as an antecedent of NL [28,43,48], and the results obtained by Gibbs et al. [43] indicated a significant positive relationship between parents’ level of NL and children’s diet quality.

Other authors of five papers referred to nutritional education interventions as a strategy to increase NL [28,48,49,50,51]. Referring to a specific type of education known as ‘edutainment’, Silk et al. [28] suggested that exposure to websites could represent a good modality to improve individuals’ NL. In their investigation, low-income European and African American mothers were exposed to nutrition education materials in three different modalities (a computer game, a website, or a pamphlet). Gibbs and Chapman-Novakofski [27] and Zoellner et al. [30] also suggested that media use may affect NL. In fact, Zoellner et al. [30] found a significant positive association between using a media channel and NL. The most frequently used media channel to obtain nutrition, food, or diet information identified by the authors of previous paper [30] was television, followed by newspapers or magazines, and Internet. Aihara and Minai [22] considered the community nexus a key factor for enhancing NL. They found that informational support and diet/nutrition information provided both from friends and health professionals had an impact on the level of NL in a wide sample of elderly people.

Some other three studies [30,34,45] adopted an individual point of view and referred to individual’s level of nutrition knowledge as a determinant factor of NL. The literature showed an association between poor NL and lower education and socio-economic level [5,22,29,45,52,53,54] according to different forms of literacy (i.e., health literacy) [55]. Finally, the authors of five papers discussed the role of age, gender, and health status as NL antecedents [5,14,22,42,54].

#### 3.3.2. Antecedents of FL

Regarding the antecedents of FL identified in the literature, the authors of eight papers referred to public health nutrition policies and health promotion interventions [24,37,56,57,58,59,60,61]. They described programs or projects that generally addressed adolescents [24,56,60,61] or children [57,59]. Other specific characteristics of these initiatives were that they were generally carried out in school settings [59,60] and involved the improvement of cooking skills [37,56,60,61]. Moreover, they also addressed disadvantaged young individuals [37,56,61]. In line with the authors that focused on NL, these researchers which discussed FL considered some social factors that affected FL: socioeconomic level [3,38,56,57,62], education, and literacy [3,23,56,62,63,64,65].

Some researchers investigating FL suggested the school as another important antecedent, considered as school environment, school setting, or school curriculum. On the one hand, Ronto et al. [62] indicated that the school curriculum could play a vital role in enhancing FL in adolescents. Furthermore, the authors of six papers emphasized the importance of teaching home economics [24,62,64,65], the vital role of academic food literacy programs [66], and the priority of school to impart food knowledge and competences [67]. On the other hand, Ronto et al. [24,64,65] emphasized the economic and human resources necessary to impart such knowledge and competences to younger people. On the other hand, Ronto et al. focused also on the school organization considering the presence and characteristics of school canteens [24,65], school kitchens [24], and the adherence to the National Healthy School Canteen Guidelines [64]. According to authors’ point of view stated in eight papers, individuals closer to the subject (i.e., parents, teachers, and peers) may represent important elements to provide specific knowledge and competences to young people [56,57,62,63,64,65,67,68]. In addition, the characteristics and factors of the home environment (i.e., family demographics) may influence individuals’ FL [62,63,64,65].

Moreover, we identified some other antecedents of FL. Godrich et al. [63] referred to the role of information coming from conventional media (television, school, magazines) and social media. Gilliland et al. [69] described a smartphone app developed for improving access to and consumption of healthy and local food. Available time [57,62,65], community nexus [65], and a closer connection to producers [63] may represent determinant factors of FL. The authors of five papers discussed the impact on FL of individuals’ age and gender [23,62], and context [3,38,60].

### 3.4. Consequences of NL and FL

#### Comparison between NL and FL Consequences

We identified and collected 53 consequences of NL and FL (See Figure 2). As seen in Figure 2, the consequences of NL identified in this literature review were also discussed by authors that focused on FL.

The authors of three papers pointed out that NL gave rise to skilled individuals [22,29,30], since the authors of other eight papers conceptualized NL as a skill-based process that led individuals to make healthy food choices and develop healthy dietary habits [22,35,45,46,47,49,51,54]. Analogously, twelve papers that discussed FL agreed that this dimension implied improved food and nutrition knowledge, and skills such as cooking [16,21,23,37,56,61,62,65,70,71,72,73]. The authors of twenty-three papers also stated that the achievement of FL leads to a healthy diet, better food and physical activity choices, and healthy behaviors (i.e., eat more vegetables or fruits, consume less soda and sweetened beverages, engage in more physical activity, increase spice use and decrease salt use, decrease serving sizes including fast food, reduce frequency of consumption of packaged and processed snacks, increase purchasing of fresh foods, and increase use of food labels when selecting foods) [3,10,11,16,20,21,22,23,24,39,40,42,56,57,59,61,62,67,68,70,71,72,73,74].

Some studies discussed the relationship between NL and the quality of diet [5,28,33,34,42,53] and some other studies confirmed this association [25,44,45]. Some publications discussed health outcomes related to NL [45,52,75,76]. An association was found between NL and lower cardiovascular disease risk [76], anthropometry measures [45], adequate lipid blood concentration [45,52], and adequate systolic blood pressure [52]. According to studies on FL, this dimension may affect obesity trends and food-related diseases in younger people [23,59].

Finally, Chang et al. [75] conceptualized consumer competences in the nutritional field as a factor affecting individuals’ food security; they identified the use of a nutritional fact panel as an effective strategy to address food insecurity. Likewise, the authors of nine papers that focused on FL conceptualized food security as a FL consequence [3,10,16,36,38,57,63,66,77]. In their opinion, the achievement of FL guarantees the access to a healthy diet. Godrich et al. [57,63] and Barbour et al. [77] explored the relationship between the two dimensions considering the Australian national context and the youngest segment of the population with low socio-economic status. They observed that food-literate individuals had the skills and confidence necessary to obtain and prepare healthy food, providing a healthier diet.

Regarding the debate on the topic of environmental sustainability, the authors of seven papers that explored FL discussed the issue [3,12,16,24,65,71,78]. From their observations, we noted that FL implies the development of critical thinking as well as awareness of the connection between the food that is consumed and the environment. Moreover, FL may favor the reduction of the human impact on the planet through cautious food choices; in fact, Ronto et al. [24] observed that dietary choices with a high content of saturated facts, sugar, and sodium were associated with the consumption of food sources. They considered the characteristics of convenience foods, which are produced spending several resources and causing a great environmental impact [79].

The development of a positive relationship with food [3,11,16,23,65], saving money, and reduced health costs [3,16,61,78] were other consequences exclusive of FL.

## 4. Discussion

To the best of our knowledge, this review is the first to identify and analyze the antecedents, components, and consequences of both FL and NL and the first attempt to collect all the definitions of the terms to propose a new one that could represent the references. Although several researchers offered a definition of the concepts, they mainly adopted a strict individual perspective. Whenever they took a population perspective, they considered it as a sum of individual behaviors, and the environment as an obstacle to adopt appropriate food choices and develop FL and NL skills. Additionally, the existing relationship between NL and FL evidenced in the literature remains to be confirmed. An exhaustive conceptualization of FL and NL that illustrates conceptual limits and meanings is still lacking. Therefore, we aimed to gain a comprehensive understanding of the topic by collecting and analyzing the characteristics of FL and NL.

The results obtained are useful to illustrate the relationship between NL and FL. In line with Krause [3], we noted that NL could be conceptualized as a specific form of FL, since both NL and FL deal with nutritional information and aim to enhance nutritional health. However, FL represents a wider concept than NL because some characteristics of the FL construct cannot be found in NL. Definitions of FL focused on individuals’ connection with food and the food system, and the objective presented in some definitions was the achievement of a healthful food system. In addition, the characteristics of FL trace a wider theme than NL, since they refer to food and the food environment. Nevertheless, the FL dimension does not incorporate NL, which could be considered as an independent concept as well. In fact, only the authors of five papers discussed both NL and FL in their papers [3,40,53,80,81]. On the contrary, several authors exclusively referred to NL and described the meaning of the concept in detail [5,13,14,16,22,27,28,29,30,31,32,35]. Moreover, only Doustmohammadian et al. [13], Guttersrud et al. [14], and Escott-Stump [31] described the nutrition environment in which individuals become able to manage their diet.

The antecedents and consequences identified contributed to tracing the relationship between the dimensions as well. The consequences of the dimensions showed that we are probably dealing with just one wide multifaceted topic that can be called ‘food and nutrition literacy’ (F&NL). In fact, the two dimensions share several consequences, and the consequences of FL constitute the major category. Furthermore, we noted that the outcomes of F&NL corresponded with the consequences of a healthy diet characterized by an abundant consumption of vegetables and that is in harmony with the ecosystem [82]. Based on the results obtained, we conclude that F&NL may strictly influence individual’s diet quality. Moreover, the results collected through this review suggested that F&NL could be described as a multidimensional concept that implies an individual dimension (knowledge, motivation, competences, and awareness) as well as the relationship between individuals and their context, aimed at consuming foods assuring nutritional health and a sustainable food system.

This definition is in accordance with the idea of public HL by Freedman [83], which opposes the individual perspective and includes health promotion and reduction of health disparities as specific goals of HL. In line with this idea, a large-scale distribution of food in line with F&NL could help to provide a healthful and sustainable diet for the population as well as a place of learning. Specifically, the reformulation of the trading system could be realized based on the approach that the achievement of a healthy diet by the population could increase everyone’s health and prosperity and respect the ecosystem that provides food [84]. In addition, some studies highlighted the effect on the diet of the environment in which the subject lives [85,86]. Food price, brand positioning, and the availability of food and foodstuffs could be important drivers for the achievement of healthful eating [87].

## 5. Limits and Perspectives

Some possible sources of bias in this review can lead to the potential exclusion of some relevant studies (e.g., we did not include any additional records recovered by the hand-search procedure). Furthermore, the search string and the selection phase of the articles based on the study objective were highly selective criteria. In addition, regarding the antecedents and consequences, we selected them considering arbitrary parameters.

Further, we did not apply the critical literacy lens [8,9] to analyze the definitions collected. This kind of analysis could be helpful to explore the F&NL construct and to underline those parts that discussed the critical analysis of food and nutrition information and the use of food in order to achieve health enhancement, and/or contribute to a more sustainable food system. Despite the importance of Nutbeam’s model [8,9], we preferred to analyze FL and NL definitions considering a content analysis with seven clusters, since it let us better identify analogies and differences between FL and NL.

Regarding the identification and collection of definitions, as well as for the antecedents and consequences of FL and NL, and the content analysis of definitions, we based this on Sørensen’s [17] methodological approach applied for the concept of HL. Nevertheless, we did not conduct a systematic review of literature given the purpose of our review and the available guidance regarding the decision to choose between a systematic review or a scoping review [88].

The antecedents, consequences, and other characteristics of the F&NL construct that are listed in this review contributed to highlight the meaning of the topic of F&NL. We also embraced a public perspective of the issue that could encourage further researches to develop and describe an executive framework in which the crucial role of the daily large-scale distribution of food in contributing to provide healthy eating for the population is considered. 

This framework should be developed with the contribution of a multidisciplinary team involving researchers as well as civil society actors. On the one hand, researchers could find adequate qualitative and quantitative methodology; on the other hand, stakeholders, managers, and other key figures could bring elements from the economic and socio-political spheres dealing with the affordability and availability of food. 

## 6. Conclusions

This review clarifies conceptual limits, shows the essence of the meaning of the terms, and proposes an integrative definition of the terms of FL and NL. The idea of F&NL described above is based on the study of a wide range of literature and it was analyzed systematically. Nevertheless, an executive framework based on F&NL still needs to be developed and described.

Individuals’ food consumption is associated with several health, environmental, and socio-economic problems. Considering the global scenario, the concepts of FL and NL emphasizing the ability of individuals to learn adequate food use seem to be insufficient to solve these problems. Nevertheless, the role of the food system in environmental and socio-economic contexts is wide and powerful. It could play a vital role in promoting health and a sustainable diet for the population or, conversely, in creating a more unsustainable world.

Several definitions proposed by previous studies allowed us to highlight the concepts of FL and NL from an individual perspective very thoroughly. Nevertheless, there is a need to advance our comprehension of FL and NL towards the construction of a comprehensive model, which embraces public perspectives and considers the decisive role of large-scale distribution of food in influencing individual’s diet quality. Our idea of F&NL emphasized the role that food system plays both on the access and the adherence to heathy eating. This conceptualization of the terms could encourage researchers to embrace a public perspective of F&NL and consequently contribute to pushing stakeholders to make decisions and take measures towards a ‘mature’ food system as an agent of sustainability in diet and nutrition.

## Figures and Tables

**Figure 1 ijerph-16-05041-f001:**
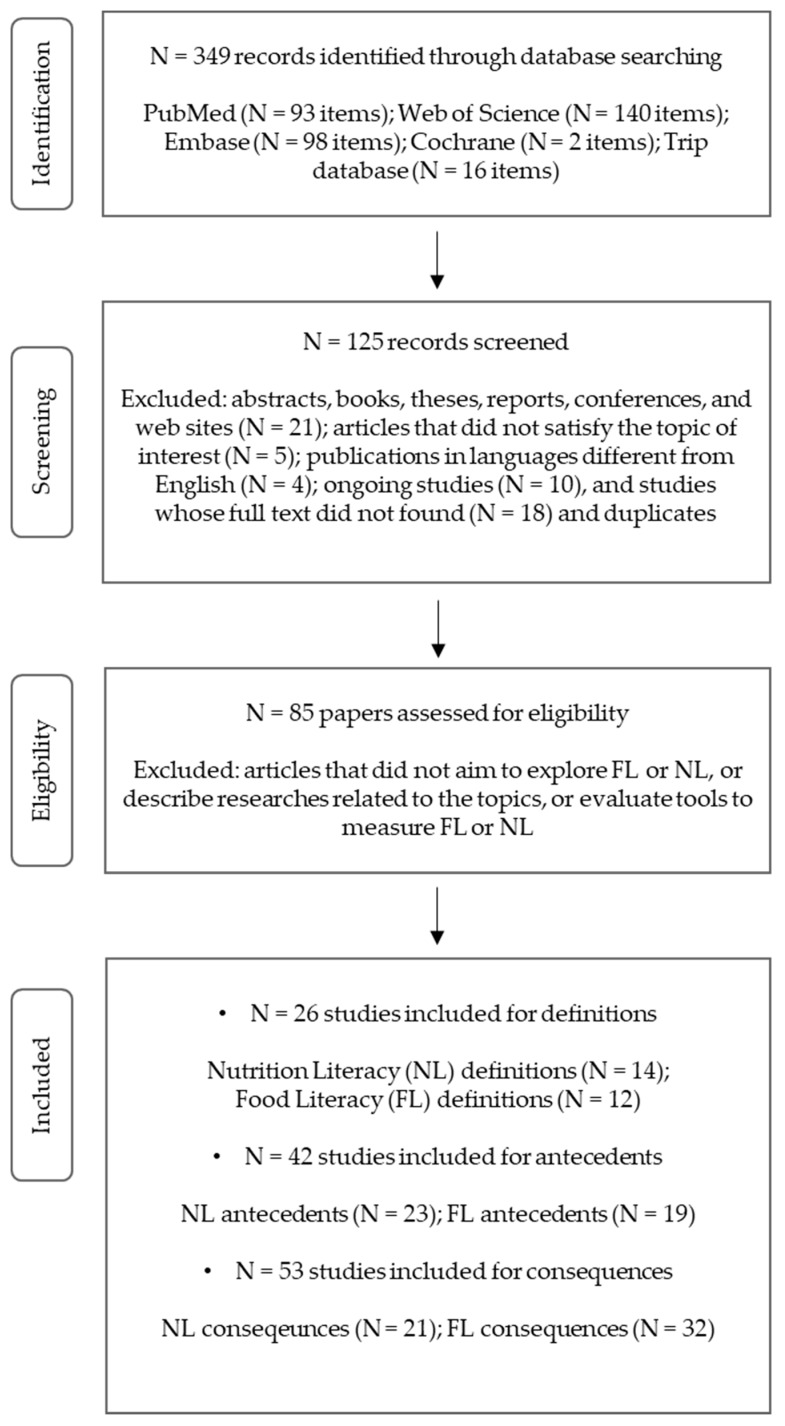
Preferred Reporting Items for Systematic Reviews (PRISMA) flow diagram for the scoping review process for the literature review of definitions, antecedents, and consequences of food literacy and nutrition literacy.

**Figure 2 ijerph-16-05041-f002:**
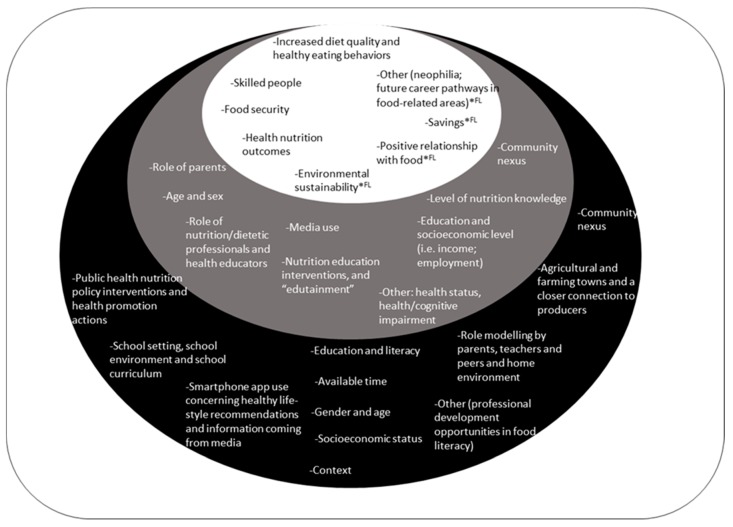
Antecedents and consequences of nutrition literacy and food literacy. The black and grey ovals indicate, respectively, the antecedents of FL and NL; the white oval shape shows the consequences of the dimensions condensed. The symbol (*) traces the consequences identified both for nutrition literacy and food literacy.

**Table 1 ijerph-16-05041-t001:** Analytic grid to conduct content analysis of definitions.

Abbreviations of Clusters	Clusters	Explication of Each Cluster
		The Part of the Sentence That…
K/C/S/A	Knowledge, competence, skills, awareness	makes explicit what literacy consists of
A	Actions	makes a list of verbs that describe cluster I
I/R	Information and resources	refers to verbs and completes the description of cluster I
S	Subject	is joined to a description by cluster I
O	Objective	describes what literacy permits to achieve
T	Time	indicates when cluster I accomplished
C	Context	describes the environment in which cluster I accomplished

**Table 2 ijerph-16-05041-t002:** Characteristics of NL and FL constructs obtained through content analysis based on seven clusters: knowledge, competence, skills, and awareness (K/C/S/A), actions (A), information and resources (I/R), subject (S), objective (O), time (T), and context (C).

Food Literacy or Nutrition Literacy	Knowledge, Competence, Skills, Awareness	Actions	Information and Resources	
**Nutrition Literacy**	-The degree to which individuals can/The capacity/The ability/Proficiency in/Dietary performance/The extent -The competence -Basic literacy skills/basic reading and writing skills -Quantitative skills -Cognitive and interpersonal communication skills -Interest in -The awareness -The knowledge -The capacity/The ability -Knowledge and skills	-To obtain/To access/How to knowledge -To find -To process -To understand/Grasp the essence/Interpret -To use/To apply -To follow -To manage -To problem solve -To read -To measure -To make right food choices -Take health-enhancing actions/Participate in actions -To interact with -Critically analyzing -To navigate -To select -To understand	-Basic nutrition information and services/Nutrition information/Simple nutrition messages/Nutrition issues/Nutrition principles/Health and nutrition concepts -Basic diet nutrition information and tools/Healthy eating information/Concepts of healthful diets/Portion size -Food labelling/Front of package/Restaurant menu labelling/Label information about fat, salt, energy, and fiber -Nutrition information guidelines/Advice -Healthy diet/Healthy-eating/Good and varied nutrition	
**Food Literacy**	-Knowledge -The motivation -The capacity/The ability -The competence -Personal skills/Practices -Decision making/Goal setting -Understanding and knowledge/The scaffolding/Understanding/Awareness/Advocating -Nutrition knowledge, skills and behaviors/Inter-related knowledge, skills, and behaviors/Food skills and practices/Broad sets of skills and knowledge -The ability/Everyday practicalities/Basic abilities related to	-To access -To use/Apply nutrition information to food choices -To understand/Evaluate -To act on that knowledge -To obtain -To interpret -To navigate/Engage/Participate -To make decisions -To understand/Understanding -To use/To select/Purchase/Prepare/Preserve food/Plan/Manage/Select/Prepare/Eat -To behave -Respecting different cultural, family, and religious belief	-Food and nutrition information and services/Relevant information/Evidence-based food and nutrition information/Food and nutrition issues -Food/Food system	
	**Subject**	**Objective**	**Time**	**Context**
**Nutrition Literacy**	-Individual -People -Client/Person that interact with nutrition counsellors/Person that interact with health professional/Students that interact with others (peers, family, and nutritionists) -From personal to social and global perspectives -Individual/s	-To make appropriate nutrition decisions -Improve quality of life/Improve one’s nutritional status and behavior/Promote healthy eating pattern -Do not be dependent on expert knowledge -To address nutritional barriers -To influence healthy eating behaviors/Maintain health and well-being	-Time spent with the expert -Daily life/Everyday life	-Nutritional environment
**Food Literacy**	-Individual/s -Youth/People -Households/Communities/Nations -Individual and collective perspective -Personal, family, and community changes	-Health enhancement/To promote nutrition goals and food well-being/To enhance nutritional health and well-being -To ensure a regular food intake that is consistent with nutrition recommendations/To meet needs and determine intake/To meet nutrition guidelines/Enhance nutritional health -To protect diet quality (dietary resilience) -Develop a positive relationship with it -Enhance physical and psychic well-being -Think critically about the relationship to -Perform actions related to -To help create behavior change -To support the achievement of personal health and a sustainable food system/Achieve health enhancement and contribute in the development of a sustainable agriculture -Concur in the accomplishment of a social equity outcomes	-Over time/Across lifespan	-Agricultural origins of foods/Food origins/Food origin and systems/Wider context of food production and nutritional health

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
