# Peer review of "Towards the Implementation of a Conceptual Framework of Food and Nutrition Literacy: Providing Healthy Eating for the Population"

_ijerph, 2019, doi:10.3390/ijerph16245041_

Round 1
Reviewer 1 Report
The manuscript titled “Towards the Implementation of a Conceptual Framework of Food and Nutrition Literacy: Providing Healthy Eating for the Population” is a well-written article about an important issue for scientific community working in nutritional epidemiology. However, there are some concerns that the authors should addressed before considering the manuscript for publication.
The authors aimed to describe and discuss evidences provided by literature in order to develop and propose a comprehensive conceptualization of food literacy (FL) and nutritional literacy (NL). Thus, they conducted a systematic review following the PRISMA criteria and the methodological approach of Sørensen et al. (2012). However, a best approach to address this kind of study would be a scoping review. The systematic reviews summarize the results of available studies (mainly controlled trials) in order to provide a high level of evidence on the effectiveness of healthcare interventions. This is the reason why they are often used as starting point for developing clinical practice guidelines. In contrast, a scoping review is a type of knowledge synthesis that follows a systematic approach to map evidence on a topic and identify main concepts, theories, sources, and knowledge gaps. Therefore, a scoping review fits better for purpose of this study. For more details, it is recommended that the authors see the PRISMA Extension for Scoping Reviews (PRISMA-ScR). A protocol and registration of the study is highly recommended. Regarding search strategy, the authors used the following combination of keywords. “food literacy” OR (“nutrition literacy” OR “nutritional literacy”) in different databases. The results retrieved in PubMed were 92 although, in the present day (18/11/2019), 148 records were identified. When did the authors perform the search strategy? The authors replicated the study by Sørensen et al. (2012) to conceptualize the terms FL and NL. Although this approach is useful and comprehensive for the synthesis of the results, the authors should also provide a critical appraisal of individual sources of evidence.Minor issues:
Page 6, line 188 (table 2): All the abbreviations used should be defined and included as footnote in this table.Author Response
Dear Reviewer,
thank you for the possibility of improving our research work.
Please find below a point-by-point response to your comments. We either uploaded all responses to reviewers' comments as a Word file.
Point 1- The authors aimed to describe and discuss evidences provided by literature in order to develop and propose a comprehensive conceptualization of food literacy (FL) and nutritional literacy (NL). Thus, they conducted a systematic review following the PRISMA criteria and the methodological approach of Sørensen et al. (2012). However, a best approach to address this kind of study would be a scoping review. The systematic reviews summarize the results of available studies (mainly controlled trials) in order to provide a high level of evidence on the effectiveness of healthcare interventions. This is the reason why they are often used as starting point for developing clinical practice guidelines. In contrast, a scoping review is a type of knowledge synthesis that follows a systematic approach to map evidence on a topic and identify main concepts, theories, sources, and knowledge gaps. Therefore, a scoping review fits better for purpose of this study. For more details, it is recommended that the authors see the PRISMA Extension for Scoping Reviews (PRISMA-ScR). A protocol and registration of the study is highly recommended.
Response: We examined the literature and discussed existing guidance regarding the decision to choose between a systematic review or scoping review based on this reviewer’s comment. Munn et al.[1] stated that the purpose of the research mainly established the decision to choose between one or the other methodological approach when synthesising evidence. The authors of previous paper[1] also suggested that the researchers might correctly decide to conduct a systematic review in case they aim to use the results of their review to answer a clinically meaningful question or provide evidence to inform practice.
Regarding the scoping review, same authors[1] suggested that this type of review still requires rigorous and transparent methods to ensure trustworthy results. Nevertheless, scoping reviews have a broader scope since generally they aim to determine what range of evidence (quantitative and/or qualitative) is available on a topic and to report this evidence[2].
Through our literature review, we attempted to grasp and synthesise the comprehensive meaning of FL and NL and their relationship by systematically identifying and collecting their definitions as well as their antecedents and consequences. We realized that both the objective and methodology of our research are in line with the guidance regarding how to conduct a scoping review.
We corrected the text and we described the summary table that we developed to record characteristics of included studies and the key information relevant to the research question.
References:
Munn, Z, Peters, MD, Stern, C, Tufanaru, C, McArthur, A, Aromataris, E. Systematic review or scoping review? Guidance for authors when choosing between a systematic or scoping review approach. BMC Medical Research Methodology 2018, 18. doi:10.1186/s12874-018-0611-x Peters, MD, Godfrey CM, Khalil H, McInerney P, Parker D, Soares CB. Guidance for conducting systematic scoping reviews. Int J Evid Based Healthc 2015, 13, 141–6. doi: 1097/XEB.0000000000000050
Location of revisions:
Section: Materials and methods (2.1. Search strategy)
Page: 2
Lines: 78-80
Section: Materials and methods (2.4. Data extraction)
Page: 4
Lines: 110-115
Point 2- Regarding search strategy, the authors used the following combination of keywords. “food literacy” OR (“nutrition literacy” OR “nutritional literacy”) in different databases. The results retrieved in PubMed were 92 although, in the present day (18/11/2019), 148 records were identified. When did the authors perform the search strategy?
Response: The search of all six databases using the search string ended in April 2018 (6/4/2018). Recently, several papers that discussed FL or NL enlarged the literature. Nevertheless, a comprehensive analysis of the concepts comparing FL with NL is still lacking. Actually, few items[1,2] were retrieved in PubMed using the search string: "food literacy" AND ("nutrition literacy" OR "nutritional literacy") (3/12/2018).
Krause’s review[1] is included in the research that we conducted. The second paper[2] is a review of the literature of existing measurement tools of FL and NL.
In our research work, we reviewed the literature considering both FL and NL in the search strategy and data analysis as well. We spent considerable time analysing data collected. Specifically, we struggled to identify and apply useful methodology to compare definitions of FL and NL and to underline analogies and differences between the concepts. We decided to specify when we performed search strategy, based on the reviewer’s comment and for the purpose of informing the reader that we searched literature until 6 April 2018. Besides, we are planning an updating of the literature reviewed.
References:
Krause, CG, Sommerhalder, K, Beer-Borst, S, Abel, T. Just a subtle difference? Findings from a systematic review on definitions of nutrition literacy and food literacy. Health Promot Int 2016, 33, 378–389. doi: 10.1093/heapro/daw084 Yuen, EYN, Thomson, M., Gardiner, H. Measuring nutrition and food literacy in adults: a systematic review and appraisal of existing measurement tools. Health Lit Res Pract 2018, 2. doi: 10.3928/24748307-20180625-01
Location of revision:
Section: Materials and Methods (2.2. Inclusion criteria)
Page:3
Line: 93
Point 3- The authors replicated the study by Sørensen et al. (2012) to conceptualize the terms FL and NL. Although this approach is useful and comprehensive for the synthesis of the results, the authors should also provide a critical appraisal of individual sources of evidence.
Response: We enlarged the paragraph of “Limits and perspectives” clarifying that regarding the identification and collection of definitions, antecedents, and consequences of FL and NL, as well as the content analysis of definitions, we based on Sørensen’s methodological approach applied for the concept of HL. Nevertheless, we did not conduct a systematic review of literature taking into account the objective of our review, which is in line with the purposes for conducting a scoping review suggested by Munn et al. (2018).
Location of revision:
Section: Limits and perspectives
Page: 14
Lines: 433-437
Point 4- Page 6, line 188 (table 2): All the abbreviations used should be defined and included as footnote in this table.
Response: Following this suggestion, we decided to reorganise Table 2 spelling out all the abbreviations. We also included in the caption of Table 2 the description of the seven clusters of the content analysis.
Location of revision:
Section: Results (3.2. Comparison between NL and FL), Table 2
Page: 7
Lines: 225-228

Reviewer 2 Report
The paper undertakes a comprehensive review of the conceptualisation of both food and nutrition/al literacy. It draws on Nutbeam's three dimensions of health literacy amongst other writings in the field to both define and explain food and nutrition(al) literacy. However what was missing was the application of the three dimensions to interrogate the data referred to in this study. There is documentation of the antecedents and consequence using existing positions but no analysis using a critical literacy lens to consider what is missing and/or still to be acknowledged.While this aspect is possibly too large a task to be included in this paper it warrants acknowledgement possibly signalling work to come.
Author Response
Dear Reviewer,
thank you for the possibility of improving our research work.
Please find below a point-by-point response to your comments. We either uploaded all responses to reviewers' comments as a Word file.
Point 1- It draws on Nutbeam's three dimensions of health literacy amongst other writings in the field to both define and explain food and nutrition(al) literacy. However what was missing was the application of the three dimensions to interrogate the data referred to in this study.
Response: At the beginning, we attempted to apply Nutbeam’s model. Specifically, we considered the three literacy levels (functional, interactive, and critical) and we reviewed each FL and NL definition traced throughout our review of the literature. We underlined and tabulated specific parts of the definition referred to each level and we summarized them. However, we realized that Nutbeam’s model did not let us trace analogies and differences between FL and NL. Consequently, we applied the content analysis of definitions and we considered seven clusters: 1. competence, skills, abilities; 2. actions; 3. information and resources; 4. objective; 5. context; 6. time; and 7. subject.
Even though we found examples in the literature of the application of Nutbeam’s model (Krause et al., 2016), we preferred to consider seven specific clusters to analyse FL and NL definitions in regard to identify analogies and differences between the concepts.
Location of the revision:
Section: Materials and Methods (2.5. Data analysis and synthesis)
Page: 4
Lines: 133-135
Point 2- There is documentation of the antecedents and consequence using existing positions but no analysis using a critical literacy lens to consider what is missing and/or still to be acknowledged.
Response: Following this comment, we enlarged the paragraph of “Limits and perspectives” stating the reason for missing the application of critical literacy lens.
Location of the revision:
Sections: Limits and perspectives
Page: 14
Lines: 426-432

Reviewer 3 Report
Overall Comments
The aim of this paper was to systematically review the literature on food literacy and nutrition literacy in order to understand how the terms have been used by researchers to date and work toward developing a comprehensive conceptualization of their usage. This is a worthwhile objective, given the frequency and diversity of usages of these concepts in food/nutrition-related research. The authors largely achieved their goal in a logical and coherent manner. Please see minor suggestions for improvement below.
Line Edits/Suggestions/Questions
Lines 30-32: I would suggest removing or relocating the first two sentences. While FL and NL are relevant in achieving a sustainable food system, they are not necessarily necessary, and may also impact other desirable outcomes.
Line 52: Can you state more explicitly if/now Nutbeam’s model will be applied to your analysis? (i.e. make it clear why are you giving it so much detail/attention in the intro).
Line 58: It feels inappropriate to state the authors’ opinion here. Moreover, I don’t the subsequent sentence is necessary. FL and NL are concepts worth exploring on their own. If you are looking for a broader implication of this analysis, I would suggest to eventually arrive at a universal/standard conceptualization of these terms that are often used somewhat ambiguously in the field of nutrition/food systems (somewhat analogous to the word “natural” on food labels).
Line 70: Please clearly define what you mean by antecedents and consequences.
Line 88: Can you provide more information about when/why texts could not be accessed? Were corresponding authors contacted in these cases?
Line 97: I think it might be helpful to refer to Appendix A here.
Line 114: Is the term “analytic grid” commonplace in content analyses?
Line 115: How were these 7 clusters of features arrived at?
Line 118-119: Please provide more detail on how the analysis was conducted. How many researchers took part in the content analysis? What was the protocol in cases where researchers disagreed?
Line 123: How were key sentences identified?
Line 130 and 149: I think each of these sections could use better transition/topic sentences. For example, I might begin each with the number of definitions retrieved and a reference to Appendix A to better orient the reader.
Table 2: This table is a bit difficult to comprehend. Perhaps spell out NL and FL in the title and also the clusters in the first column (otherwise the reader has to refer back to Table 1, which is cumbersome). I’m still not entirely sure what I am supposed to be taking away from the table. Is the content listed in any particular order? Also, why are the constructs presented in this order instead of the order presented in table 1?
Sections 3.2.1 – 3.2.4: It would be helpful to include the acronyms listed in the left column of table 2 for ease of linking the two. I’m also not sure four separate sections are needed.
Figure 2: It would be helpful to add a color-coded key to this figure, to indicate what the black, grey, and white ovals indicate.
Line 354: This conclusion feels like a stretch. I would suggest eliminating it, since the evidence presented does not clearly substantiate the conclusion. Softer language could be used to suggest several possible applications of your findings instead.
Line 387: I believe there is a typo here. Still needs?
Additional note: The authors (of this paper) often refer to the authors of other papers as “authors”. It would be more appropriate to either refer to previous literature by the first author’s last name (or generically as “one previous paper …”) or specify the number of papers (e.g. “six previous papers observed …” for line 237) in the case where multiple papers are references. Using the words “some others” or “other authors” sounds vague, and there’s no reason not to list the number.
Author Response
Dear Reviewer,
thank you for the possibility of improving our research work.
Please find below a point-by-point response to your comments. We either uploaded all responses to reviewers' comments as a Word file.
1- Lines 30-32: I would suggest removing or relocating the first two sentences. While FL and NL are relevant in achieving a sustainable food system, they are not necessarily necessary, and may also impact other desirable outcomes.
Response: Following the suggestion, we relocated the first two sentences.
Location of revision:
Section: Introduction
Page: 1
Lines: 30-32
2- Line 52: Can you state more explicitly if/now Nutbeam’s model will be applied to your analysis? (i.e. make it clear why are you giving it so much detail/attention in the intro).
Response: In the introduction, we emphasised Nutbeam’s model, taking into consideration that several authors developed their definitions of FL and NL referring to it. We briefly described it with the aim to synthesise the characteristics of the model that recur in some FL and NL definitions discussed in the rest of the paper. Krause et al. (2016) referred to Nubeam’s model to study and compare FL and NL definitions, however we decided to not apply it and use the content analysis of definitions. We intended to trace analogies and differences between FL and NL and we could not fulfill this task with Nutbeam’s model.
Location of revision:
Section: Materials and Methods (2.5. Data analysis and synthesis)
Page: 4
Lines: 133-135
3- Line 58: It feels inappropriate to state the authors’ opinion here. Moreover, I don’t the subsequent sentence is necessary. FL and NL are concepts worth exploring on their own. If you are looking for a broader implication of this analysis, I would suggest to eventually arrive at a universal/standard conceptualization of these terms that are often used somewhat ambiguously in the field of nutrition/food systems (somewhat analogous to the word “natural” on food labels).
Response: The line 58 referred to seven previous papers (Vidgen & Gallegos, 2014; Block et al., 2011; Cullen et al., 2015; Doustmohammadian, et al. 2017; Guttersrud et al., 2013; Krause et al., 2018; Palumbo, 2016). We summarised their opinions taking into consideration their conceptualization of the terms of FL and NL. For example, Cullen et al. (2015) declared that “food [is] the ability to make decisions to support the achievement of personal health and a sustainable food system” and Block et al. (2011) stated “food literacy entails […] acting on that knowledge in ways consistent with promoting nutrition goals and FWB [food well-being]”. Following the reviewer’s suggestion, we reformulated the sentence of line 58 and we removed lines 59-61.
Location of revision:
Section: Introduction
Page: 2
Lines: 58-59
4- Line 70: Please clearly define what you mean by antecedents and consequences.
Response: Regarding the antecedents, we mean distal (socio-economic conditions, cultural, and environmental conditions, education, living and working conditions, housing) as well as more proximal (age, sex, and general literacy) factors which could potentially influence individuals’ FL and NL. Regarding the consequences, we reviewed papers considering health related outcomes (i.e. body mass index, lipid blood concentration), and other factors that may impact individuals’ health (i.e. food safety and food security). We considered this methodological approach in line with Sørensen et al. (2012).
Location of revision:
Section: Materials and Methods
Page: 2
Lines: 70-76
5- Line 88: Can you provide more information about when/why texts could not be accessed? Were corresponding authors contacted in these cases?
Response: In some cases, neither full-text nor abstract was found in the database or Google Scholar, or Google. We did not contact the authors.
Location of revision:
Section: Materials and Methods (2.2. Inclusion criteria)
Page: 4
Lines: 97-98
6- Line 97: I think it might be helpful to refer to Appendix A here.
Response: Following the suggestion, we added the reference to Appendix A in the paragraph “Data extraction”.
Location of revision:
Section: Materials and Methods (2.4. Data extraction)
Page: 4
Lines: 116-117
7- Line 114: Is the term “analytic grid” commonplace in content analyses?
Response: The literature provides example. Krause et al. (2016) developed a table called “analytic grid” summarizing characteristics of functional, interactive, and critical health literacy to review FL and NL definitions.
8- Line 115: How were these 7 clusters of features arrived at?
Response: Sørensen et al. identified six clusters in their analysis of HL definitions. We considered the same clusters (1. competence, skills, abilities; 2. actions; 3. information and resources; 4. objective; 5. context; and 6. time) and we added the 7th cluster: 7. subject. We considered one more cluster with the aim to distinguish definitions that adopted individual perspective from those referred to a wider perspective. Definitions focused on individual perspective generally refer to a person, or individual,…. In the other case, the terms of population, or community,… were used in the definitions.
Location of revision: Section: Materials and Methods (2.5. Data analysis and synthesis)
Page: 4
Lines: 138-141
9- Line 118-119: Please provide more detail on how the analysis was conducted. How many researchers took part in the content analysis? What was the protocol in cases where researchers disagreed?
Response: The systematic search of the literature using the search string ‘food literacy’ OR (‘nutrition literacy’ OR ‘nutritional literacy’) was independently performed by two authors (V.V. and C.M.). The same authors (V.V. and C.M.) screened the full text of 85 papers and collected definitions, antecedents, and consequences. We also extracted, summarised, and tabulated following key information from each publication: title of the publication, author(s), publication year, abstract, definitions of FL or NL, antecedents, and consequences. Regarding the content analysis of the definitions, one researcher (V.V.) examined FL and NL definitions according to each cluster. Sentences (or part of sentences) of the definitions related to the clusters were underlined and tabulated. Basing on the summary table, two reviewers (G.B. and C.L.) independently reviewed the analysis of the definitions. In cases where the reviewers disagreed, the content analysis of the definition was repeated and submitted to the reviewers again.
Location of revisions:
Section: Materials and Methods (2.3. Study selection)
Page: 4
Lines: 105-108
Section: Materials and Methods (2.5. Data analysis and synthesis)
Page: 5
Lines: 145-149
10- Line 123: How were key sentences identified?
Response: Basing on the meaning of antecedents and consequences of FL and NL, two authors (V.V. and C.M.) reviewed papers and underlined the sentences reporting antecedents and/or consequences. Secondly, all sentences were tabulated in an electronic spreadsheet, in which other key information from each publication were collected (title of the publication, author(s), publication year, abstract, definitions of FL or NL).
Location of revision:
Section: Materials and Methods (2.5. Data analysis and synthesis)
Page: 5
Lines: 152-156
11- Line 130 and 149: I think each of these sections could use better transition/topic sentences. For example, I might begin each with the number of definitions retrieved and a reference to Appendix A to better orient the reader.
Response: Following the suggestion, we added topic sentences at the top of the sections. We briefly summarised the results described in the rest of the paragraph and we also stated the number of definitions collected. We added the reference to Appendix A.
Location of revisions:
Section: Results (3.1.1. Definitions of NL)
Page: 5
Lines: 160-161
Section: Results (3.1.2. Definitions of FL)
Page: 6
Lines 184-186
12- Table 2: This table is a bit difficult to comprehend. Perhaps spell out NL and FL in the title and also the clusters in the first column (otherwise the reader has to refer back to Table 1, which is cumbersome). I’m still not entirely sure what I am supposed to be taking away from the table. Is the content listed in any particular order? Also, why are the constructs presented in this order instead of the order presented in table 1?
Response: Following the suggestion, we modified Table 2 spelling out NL and FL and the clusters. We also reorganised the clusters in the same order of Table 1 and we realized that horizontal direction for these pages could be a better choice. We decided to maintain the table in the paper taking into account it showed the content analysis of definitions.
Location of revision:
Section: Results (3.2. Comparison between NL and FL), Table 2
Page: 7
Lines: 225-228
13- Sections 3.2.1 – 3.2.4: It would be helpful to include the acronyms listed in the left column of table 2 for ease of linking the two. I’m also not sure four separate sections are needed.
Response: We included the acronyms and we removed four separate sections to better orient the reader as suggested by the reviewer.
Location of revision:
Section: Results (3.2. Comparison between NL and FL)
Page: 10
Lines: 229, 239, 249, 258, 265
14- Figure 2: It would be helpful to add a color-coded key to this figure, to indicate what the black, grey, and white ovals indicate.
Response: Following the suggestion, we added a color-coded in the caption of the figure and we removed the description of the figure in the paragraph.
Location of revision: Section: Results (3.3. Antecedents of FL and NL) Figure 2
Page: 11
Lines: 276-278
15- Line 354: This conclusion feels like a stretch. I would suggest eliminating it, since the evidence presented does not clearly substantiate the conclusion. Softer language could be used to suggest several possible applications of your findings instead.
Response: To address the concern, we eliminated this conclusion and we reformulated the sentence using a softer language.
Location of revision:
Section: Discussion
Page: 14
Lines: 406-407
16- Line 387: I believe there is a typo here. Still needs?
Response: The typo in this line was corrected.
Location of revision:
Section: Conclusions
Page: 15
Line: 452
17- Additional note: The authors (of this paper) often refer to the authors of other papers as “authors”. It would be more appropriate to either refer to previous literature by the first author’s last name (or generically as “one previous paper …”) or specify the number of papers (e.g. “six previous papers observed …” for line 237) in the case where multiple papers are references. Using the words “some others” or “other authors” sounds vague, and there’s no reason not to list the number.
Response: Following the suggestion, we modified the words “some authors”, “some other authors”, and “other authors”. We reviewed the full paper, and we referred to previous literature by the first author’s last name (e.g. “Doustmohammadian et al.[13] and Guttersrud et al.[14]” (Page: 5, Line: 171)), or specify the number of papers (i.e. “five papers” (Page: 5, Line: 162), “three papers” (Page: 5, Line: 165)), or with the words “previous papers” (e.g. “Five previous papers” (Page: 6, Lines: 207-208)).

Round 2
Reviewer 1 Report
The authors have provided adequate responses to the reviewer's concerns. The paper has improved considerably. All the issues required have been addressed properly. No more changes are required.